# Optimizing Rhabdomyosarcoma Treatment in Adolescents and Young Adults

**DOI:** 10.3390/cancers14092270

**Published:** 2022-05-02

**Authors:** Atsushi Makimoto

**Affiliations:** 1Department of Laboratory Medicine, Tokyo Metropolitan Children’s Medical Center, 2-8-29, Musashidai, Fuchu 183-8561, Tokyo, Japan; atsushi_makimoto@tmhp.jp; Tel.: +81-42-300-5111 (ext. 5177); 2Department of Hematology/Oncology, Tokyo Metropolitan Children’s Medical Center, 2-8-29, Musashidai, Fuchu 183-8561, Tokyo, Japan

**Keywords:** rhabdomyosarcoma, adolescent and young adult, AYA, multimodal treatment, chemotherapy, neoadjuvant, adjuvant, surgery, radiation therapy

## Abstract

**Simple Summary:**

Due to their rarity, we still know comparatively little about rhabdomyosarcomas (RMS) in adolescents and young adults (AYA). Factors responsible for the significantly poor outcomes in AYA-RMS are tumor biology, physiological factors specific to the age group concerned, refractoriness to multimodal treatments, and various psychosocial and medical care issues. The present review aims to examine these issues and offers possible solutions based on integrating new developments and findings with older, fundamental evidence regarding pediatric RMS.

**Abstract:**

Rhabdomyosarcoma (RMS) is the most common form of soft tissue sarcoma in children, but can also develop in adolescents and young adults (AYA). The mainstay of treatment is multi-agent chemotherapy, ideally with concomitant local treatment, including surgical resection and/or radiation therapy. Although most treatment decisions for RMS in AYA are based on scientific evidence accumulated through clinical studies of pediatric RMS, treatment outcomes are significantly inferior in AYA patients than in children. Factors responsible for the significantly poor outcomes in AYA are tumor biology, the physiology specific to the age group concerned, refractoriness to multimodal treatments, and various psychosocial and medical care issues. The present review aims to examine the various issues involved in the treatment and care of AYA patients with RMS, discuss possible solutions, and provide an overview of the literature on the topic with several observations from the author’s own experience. Clinical trials for RMS in AYA are the best way to develop an optimal treatment. However, a well-designed clinical trial requires a great deal of time and resources, especially when targeting such a rare population. Until clinical trials are designed and implemented, and their findings duly analyzed, we must provide the best possible practice for RMS treatment in AYA patients based on our own expertise in manipulating the dosage schedules of various chemotherapeutic agents and administering local treatments in a manner appropriate for each patient. Precision medicine based on state-of-the-art cancer genomics will also form an integral part of this personalized approach. In the current situation, the only way to realize such a holistic treatment approach is to integrate new developments and findings, such as gene-based diagnostics and treatments, with older, fundamental evidence that can be selectively applied to individual cases.

## 1. Introduction

RMS is the most common form of soft tissue sarcoma in children and has an annual incidence of 4.9 per one million children under the age of 19 years [1]. Although RMS can develop in adults, most cases are diagnosed in patients younger than 20 years, according to the Surveillance, Epidemiology and End Results (SEER) program [2]. In the present review, the author focuses on RMS in adolescent and young adult (AYA) patients, examines the various issues that can arise in their treatment and care, discusses possible solutions, and provides an overview of the literature on the topic with some observations from his own experience.

Pediatric oncologists are well aware that RMS outcomes worsen with age. Joshi et al. analyzed data on patients older than 10 years with a significantly poorer failure-free survival (FFS) rate than children aged 1 to 9 years (51 vs. 72%) in the Intergroup Rhabdomyosarcoma Study (IRS) III, IV pilot, and IV and concluded that age is an independent prognostic factor in pediatric RMS [3]. Sultan et al. analyzed data from the SEER database containing 1071 adults (>19 years of age) and 1529 children (= or <19 years of age) with a diagnosis of RMS between 1973 and 2005 and found that adult patients with RMS had a significantly worse overall survival (OS) rate (27%) than children (61%) [4]. EUROCARE-5, an epidemiological study, reported a five-year OS rate of 66.6% among children aged 0–14 years in contrast to 39.6% for adolescents aged 15–19 years [5].

A number of studies in various, related fields universally concluded that the reason for the poor RMS outcomes in AYA patients was multifactorial [6]. Figure 1 summarizes the factors significantly influencing the poor outcomes and their relationship to each other. Internal factors included tumor biology and physiology specific to the patients’ age group, while external factors included multimodal treatments, the combinations used, and various psychosocial and medical care issues.

## 2. Tumor Biology of RMS

RMS refers to malignant tumors with skeletal muscle differentiation. Although all RMS subtypes share this morphological feature, they are otherwise highly heterogenous, with each subtype possessing its own distinct morphological, molecular, and clinical features. RMS was first discovered in 1854 and named sarcoma botryoides [7]. The discovery of embryonal RMS (ERMS) occurred in 1894 [8], followed in 1956 by the discovery of alveolar RMS (ARMS) [9]. Since then, numerous modifications to the diagnostic criteria and classification of RMS have been made, with the most recent revision of RMS classification made by the World Health Organization (WHO) in 2020 [10]. The WHO criteria describe four RMS subtypes, namely, ERMS, ARMS, spindle cell/sclerosing RMS (SSRMS), and pleomorphic RMS (PRMS). Since these histological subtypes, together with molecular/genetic abnormalities of the tumor cells, are crucial determinants of the clinical behavior of RMS, a precise diagnosis of each case and an appropriate treatment strategy for the relevant subtype are necessary.

In pediatric oncology, ERMS and ARMS are two major variants with a characteristic, histological appearance, as well as distinctive, molecular/genetic abnormalities. Although the clinical behavior and outcomes differ significantly among these subtypes, both share the same chemotherapeutic regimens. The standard chemotherapy regimen of the Children’s Oncology Group (COG) in North America consists of vincristine, actinomycin D, and cyclophosphamide (VAC) [11,12,13,14,15,16], whereas the regimen of the European Pediatric soft tissue sarcoma Study Group (EpSSG) uses ifosfamide instead of cyclophosphamide (IVA) [17,18]. ARMS has a significantly worse outcome than ERMS [14]. Although the two subtypes share the requirement for surgical intervention, the radiotherapy dosage tends to be higher in ARMS owing to its greater aggressiveness [15]. Most cases of ARMS are fusion-positive (FP), meaning that the RMS involves a *PAX3/7-FOXO1* genomic rearrangement. The presence of *FOXO1* fusion genes is also a strong predictor of a poor outcome [19]. On the other hand, in pediatric oncology, ERMS is often thought of as identical with fusion-negative (FN) RMS. Some FN-RMS tumors carry point mutations in the RAS pathway genes (predominantly in infants) or in *FGFR4*, *PIK3CA*, *TP53*, or *MYOD1* [20]. Although the prognostic value of the RAS mutations is unclear, *MYOD1* (closely related to SSRMS, described below) and *TP53* are predictors of a poor prognosis in patients with RMS [21,22]. 

SSRMS is a FN-RMS subtype with a fascicular spindle cell and/or sclerosing morphology and accounts for 3–10% of RMS cases [10]. To date, three subtypes are known, including congenital SSRMS with *VGLL2*/*NCOA2*/*CITED2* rearrangements, *MYOD1*-mutant SSRMS, and intraosseous SSRMS with a *TFCP2*/*NCOA2* rearrangement. The prognosis of *MYOD1*-mutant SSRMS is equally poor in pediatric and adult patients. According to a case series published by the Memorial Sloan Kettering Cancer Center (MSKCC) in which 16 of the 30 patients were older than 16 years, the 3-year and 4-year overall survival rate was 36% and 18%, respectively [23].

PRMS, a high-grade, pleomorphic tumor with a skeletal muscle phenotype, is also a FN-RMS subtype and primarily affects adult patients. Furlong et al. [24] reported 38 patients with PRMS with the median age of 51 years (range: 21 to 81 years). Follow-up data on 30 (79%) of these patients revealed that 70% died of the disease (mean survival: 20 months; range: 1 to 108 months), 3% were still alive with the disease at 12 months (*n* = 1), and 27% apparently experienced a complete resolution (mean time to resolution: 83 months; range: 18 to 108 months). Since PRMS is usually treated in a similar way to non-RMS soft tissue sarcoma in adults, PRMS should be carefully differentiated when deciding on a treatment for AYA RMS.

The poor prognosis of AYA RMS may be partly explained by differences in biology. According to a study by the Associazione Italiana di Ematologiae Oncologia Pediatrica (AIEOP), which compared 567 children (aged 0 to 14 years) and 76 adolescents (aged 15 to 19 years) with RMS who underwent four consecutive clinical trials, AYA RMS had more alveolar subtypes (47.4% vs. 32.6%), lymph node infiltration (39.1% vs. 23.3%), and metastases at diagnosis (30.7% vs. 17.8%) [25]. According to a report by Sultan et al., tumors in adult patients were more likely to occur in an unfavorable site and included more instances of PRMS and RMS not otherwise specified (RMS-NOS) [4]. AYA RMS, which theoretically falls between pediatric and adult RMS, includes subgroups with a poor prognosis, such as SSRMS, PRMS, and RMS-NOS. Moreover, a genomic characterization study of MSKCC (MSK-IMPACT) revealed that *TP53* mutations, which may predict poor prognosis of RMS, were more common in older patients (median age 28.3 years in patients with *TP53* mutations vs. 14.6 years in those with *TP53* wild type; *p* = 0.05) [22]. Large genomic studies in the future may help to describe the genetic subtypes in AYA-RMS and improve treatment strategies based on the genomic findings.

## 3. Physiology of AYA

AYA patients, especially adolescents, experience numerous psychological and physical changes that can affect treatment. These include increases in height and weight, body composition, hormonal environment, organogenesis, changeable body mass index, and compliance with treatment [26]. Moreover, the pharmacokinetics (PK) of anticancer drugs vary widely among individuals. For example, the clearance rate of vincristine normalized to body weight was significantly faster in children than adolescents in one study [27], whereas another study reported no relationship between age and pharmacokinetics [28]. Actinomycin D tends to have higher area under the curve (AUC) in children with a body weight <40 kg than in adolescents [29]. Although there are very few data on cyclophosphamide PK in AYA patients, the drug reportedly has a shorter half-life and higher clearance rate in children than in adults [30]. Moreover, the interpatient variability is considerable and is accompanied by a certain level of age-dependency because cyclophosphamide is a prodrug that undergoes a complicated process of metabolic activation and interaction [31].

Pharmacodynamic studies have reported reproducible patterns of age-related toxicity from VAC chemotherapy and related regimens. In IRS-IV, in which patients were randomly assigned to one of three different chemotherapy regimens (VAC: vincristine, actinomycin D, and cyclophosphamide; VAI: vincristine, actinomycin D, and ifosfamide; VIE: vincristine, ifosfamide, and etoposide), patients aged 15 to 20 years experienced less neutropenia (odds ratio [OR]: 0.43; *p* < 0.0001), thrombocytopenia (OR: 0.41; *p* < 0.0001), anemia (OR: 0.34; *p* < 0.0001), and infection (OR: 0.41; *p* < 0.0001) than younger patients, despite receiving similar amounts of chemotherapy. In contrast, peripheral nervous system toxicity was higher in adolescents aged > 10 years (OR: 4.18; *p* < 0.0001) [32]. Although the outcomes were inferior in adolescents (an estimated five-year event-free survival rate of 58% for patients aged 15–20 years vs. 75% for all patients), it was unclear whether the difference in toxicity had any impact on the outcomes. A subsequent study using a patient cohort registered in COG-D9803, which randomly assigned patients to two different chemotherapy regimens (VAC vs. alternating VAC plus topotecan, cyclophosphamide), also produced very similar results [33].

Analyses of patients with metastatic RMS registered in COG-ARST0431 [34], in which intensive, multi-agent (six drugs) chemotherapy, including a dose-compressed cycle of ifosfamide/etoposide (IE), vincristine/doxorubicin/cyclophosphamide (VDC), and vincristine and irinotecan (VI) was administered, showed that adolescents (aged > 13 years [*n* = 59]) were more likely to have nausea (17 vs. 4%, *p* = 0.06) and pain (20 vs. 6%) than children (aged 13 years or less [*n* = 49]) in the early phase of treatment. Adolescents were less likely to complete therapy (63 vs. 76%) and more likely to have unplanned dose modifications outside the protocol (23 vs. 2.7%). The three-year event-free survival (EFS) rate was 26% for adolescents compared with 46% for children. Although the relationship among toxicity, treatment compliance, and outcomes was unclear, the dose-intensity of chemotherapy tended to be lower in adolescent patients with RMS, at least in the study setting.

The trend of decreasing dose-intensity in VAC chemotherapy for AYA RMS was also evident in our monocentric case-series enrolling 27 children and 18 adults [35]. The relative dose-intensity of vincristine and cyclophosphamide tended to decrease as the patients became older (Figure 2). The mean relative dose-intensity decreased significantly in adults in the late phase of the treatment (Table 1). VAC chemotherapy using high dose cyclophosphamide (up to 2.2 g/m^2^) does not seem viable for AYA RMS, not only because of intolerance to the treatment, but also due to the risk of infertility in this age group. Minimizing anxiety about the treatment outcomes, including adverse events, is very important for psychosocially vulnerable AYA patients. The following section will discuss ways of optimizing the chemotherapy regimen.

The mean relative dose-intensity was significantly lower in adults in the late phase of the treatment course. This table was used with the permission of Kojima et al. [35].

## 4. Development of Optimal Multimodal Treatment for AYA RMS

The mainstay of RMS treatment is chemotherapy because RMS is a markedly chemoresponsive disease. As mentioned above, the standard treatment is the VAC regimen in North America [11,12,13,14,15,16] and the IVA regimen in Europe [17,18]. Other chemotherapy regimens that are available for patients with newly diagnosed RMS are summarized in Table 2. The response rate to these types of chemotherapy for ERMS and ARMS is reportedly 70–80% [13,36]. On the other hand, RMS cannot be cured only with chemotherapy, despite the high response rate. Local control measures, including surgery and/or radiotherapy, need to be added for complete disease control at the original tumor site.

Neoadjuvant chemotherapy, which is administered prior to surgical resection, mainly aims to improve local tumor control in relation to local treatment modalities. Its effectiveness is evaluated using the response rate as measured by tumor shrinkage, although this may not constitute sufficient evidence of “good” treatment leading to a cure. In contrast, adjuvant chemotherapy, which specifically targets minimal residual disease after surgical resection, aims to prevent metastatic (and partly, local) recurrence by eliminating systemic micrometastatic tumors. According to the Norton–Simon (NS) hypothesis, chemotherapeutic agents are most effective in this setting [40]. A certain dosage and duration of adjuvant chemotherapy is necessary to eradicate residual disease in RMS patients even when the best possible combination of agents is used. According to the Goldie–Coldman (GC) hypothesis, the chance of a cure should be maximized by deploying effective multi-agent chemotherapy prior to the development of drug-resistant cells through random genetic events [41].

Figure 3 shows the basic structure of multimodal therapy for RMS and the key clinical questions. Although each clinical question should be answered by conducting a well-designed clinical trial, this requires considerable time (up to 5–6 years per trial) because of the rarity of RMS, as well as issues with financing and the effort of coordinating national and international institutions with different policies. In cases of AYA RMS, in particular, it is almost impossible to enroll a sufficient number of patients to ensure the requisite statistical power for a clinical trial. Moreover, it is becoming increasingly difficult for a treatment for RMS as a single entity to be tested in a clinical trial due to accumulating evidence on tumor genomics and subgrouping, as discussed above.

Given these challenges, recourse may be had to the significant amount of data inherited from past studies, which can be taken on board, together with more recent data and our own experiences, to create fresh perspectives on the issue. Conducting a clinical trial is as demanding as planning one. Considerable time is also required to establish a protocol for a well-designed clinical trial. Meanwhile, the best practices for AYA RMS treatment must be provided using our own expertise in manipulating dosage schedules of chemotherapy agents and administering local treatments in a manner optimized for individual patients. The following discussion will hopefully aid readers in optimizing treatments for not only patients with AYA RMS, but also childhood RMS.

### 4.1. Differentiating Neoadjuvant and Adjuvant Chemotherapy

Although the excellent chemoresponsiveness of RMS is an advantage in treatment, it may lead to overestimating the effectiveness of systemic chemotherapy and misjudging the role of neoadjuvant and adjuvant chemotherapy. The history of VAC chemotherapy shows that neoadjuvant and adjuvant chemotherapy were clearly different, especially in clinical groups III and IV RMS and IRS I-III [11,12,13]. IRS later apparently administered a uniform chemotherapy regimen before and after local control therapy without differentiating between neoadjuvant and adjuvant chemotherapy [15,16].

IRS-III [13] compared the response rate in three different regimens (#36: pulsed VDC + cisplatin + etoposide; #35: #36 without etoposide; #34: pulsed VAC) at induction therapy completion (week 20) and later. Although the complete remission rate was better for regimen #36 (48% at 20 weeks and 84% in later phases) than for regimen #35 (45% and 78%) or #34 (39% and 79%) in the group III patients, the five-year progression-free survival (PFS) and overall survival (OS) rates did not differ. The group IV patients, who showed the same trends in the three regimens, will be discussed later. The clinical group III patients in IRS-IV showed no correlation between the early response and long-term survival rates [36].

Weichteilsarkom Studiengruppe (CWS)-91 investigated the effectiveness of intensive five-drug combination EVAIA (etoposide, vincristine, actinomycin-D, ifosfamide, doxorubicin) as an induction therapy for group III patients with RMS and related sarcoma, but failed to show superiority to VAIA (EVAIA without etoposide) in CWS-86 [42]. CWS also retrospectively analyzed the relationship between the initial response and outcomes in 529 patients under 21 years of age with localized RMS. Although five-year overall survival in the stable/progressive disease group was significantly poorer than in responsive patients, there was no significant difference between partial responders (>33% reduction of tumor size) and objective responders (0–33% reduction of tumor size) [43].

These data may have discouraged researchers from trying to optimize neoadjuvant chemotherapy to improve tumor resectability and the local control rate. Recent studies of RMS in North America have sought to evaluate the effect of adding a single, novel agent or a doublet combination either as a substitute for the standard VAC or for alternation with the standard VAC cycles [15,16,37,38]. These studies generally used the same regimens regardless of the treatment phase (induction or continuation).

However, devising superior drug combinations and treatment schedules for neoadjuvant chemotherapy may contribute to improving tumor resectability and the local control rate. In addition, superior drug combinations and dosage schedules for reducing systemic minimal residual disease should be feasible with adjuvant chemotherapy. Clinical trials of RMS treatments, except EpSSG [17,18] and CWS studies [44] in which vincristine was administered weekly if only in the preoperative phase of chemotherapy, have yet to assess the effect of differentiating these two types of chemotherapy. The weekly administration of vincristine is effective in stemming RMS at its onset, but is theoretically less advantageous as adjuvant chemotherapy after tumor resection. The continuation phase of treatment without weekly vincristine to avoid excessive neurotoxicity is one way of optimizing AYA RMS treatment.

### 4.2. Optimal Dosing of Alkylating Agents

IRS-IV, in which three different regimens (VAC, VAI, and VIE) were administered for 28 weeks (the cumulative ifosfamide dosage was capped out of concern for nephrotoxicity), followed by VAC until week 46 failed to prove the superiority of VAI or VIE in terms of the endpoint of FFS (75% in VAC, 77% in both VAI and VIE) [14,39]. Conversely, VAI and VIE might be as effective as VAC for most patients with RMS. In this regard, the IVA regimen with ifosfamide 6 g/m^2^ used in European studies [17,18] should be regarded as a standard regimen together with the VAC regimen in North America. IRS-IV [14,39], COG-D9802 [45], and D9803 [15] used an escalated dosage (2.2 g/m^2^) of cyclophosphamide equivalent to the standard dose (9 g/m^2^ in 5 days) of ifosfamide and demonstrated a survival benefit in patients with local or regional ERMS [46]. Although group III patients with an unfavorable tumor site showed little improvement in outcomes after treatment with dose-escalated cyclophosphamide, this intensified VAC regimen is a possible option for intensifying basic chemotherapy for ERMS treatment.

COG-D9803 [15], which used a new treatment regimen consisting of vincristine, topotecan, and cyclophosphamide (VTC), showed a similar FFS rate for VAC (73%) and alternating VAC/VTC (68%) for intermediate-risk RMS. The total cyclophosphamide dose in the VAC regimen was 30.8 g/m^2^ compared to 25.1 g/m^2^ (20% reduction) in the VAC/VTC regimen. ARST0531 [16], which was subsequently performed for intermediate-risk RMS, used reduced-dose (1.2 g/m^2^) cyclophosphamide in the VAC (currently “standard” or “intermediate dose” VAC) regimen and compared the outcomes of the treatments with the VAC and alternating VAC/VI regimens. The total cyclophosphamide dosage in the VAC regimen was 16.8 g/m^2^ compared with 8.4 g/m^2^ (50% reduction) in the VAC/VI regimen. Although the four-year EFS rate was 63% for VAC and 59% for VAC/VI (*p* = 0.51), the local failure rate was significantly higher than for D9803 (27.9% vs. 19.4%; *p* = 0.03) [45]. The reason for the high local failure rate might possibly be the reduction of the cyclophosphamide dosage.

Reducing the dose of alkylating agents decreases toxicity to the gonadal functions and is recommended for AYA RMS, but should be weighed against an increased risk of local failure. Determining the appropriate cyclophosphamide and ifosfamide dosage in the VAC and IVA regimen, respectively, is challenging, but the optimal range is 1.2 to 2.2 g/m^2^/course for cyclophosphamide and 6 to 9 g/m^2^/course for ifosfamide. If the emphasis in treatment falls on local control, using higher doses of alkylating agents, which are usually administered in the neo-adjuvant or induction chemotherapy phase, may be considered.

### 4.3. Optimal Chemotherapy Duration

Treatment duration importantly influences the treatment burden and the patient’s quality of life. Clinical trials in North America have progressively shortened, from longer trials lasting 104 weeks, such as IRS I-III, to shorter trials lasting 42 weeks for higher-risk patients. Under the EpSSG, the trial length increased from 27 weeks to 51 weeks as a result of adding maintenance chemotherapy, consisting of vinorelbine plus oral cyclophosphamide for 24 weeks [46], especially in patients with a high risk of RMS. Although the optimal duration of chemotherapy differs according to the risk factors in each patient, some continuation maintenance chemotherapy (adjuvant chemotherapy) is needed to reduce residual disease in RMS.

The RMS2005 trial conducted by the EpSSG demonstrated a benefit of adding a 24-week maintenance chemotherapy regimen, consisting of vinorelbine and oral cyclophosphamide, in terms of OS for non-metastatic high-risk RMS [18]. Among 371 patients enrolled, the five-year disease-free survival rate was 77.6% with maintenance chemotherapy versus 69.8% without maintenance chemotherapy (hazard ratio [HR]: 0.68; *p* = 0.061), and the five-year overall survival rate was 86.5% vs. 73.7% (HR: 0·52; *p* = 0.0097). Although the results are very promising, several issues in standardizing maintenance chemotherapy for AYA RMS must be dealt with. First, the efficacy of maintenance chemotherapy was demonstrated in specific populations, such as patients with non-metastatic, high-risk RMS. Second, adding the 24-week maintenance chemotherapy might only be maximally effective in conjunction with the relatively short, 27-week, EpSSG IVA regimen; its effectiveness when combined with the 42-week North American VAC regimen is unclear. Last, the long treatment duration in the outpatient setting is not suitable for AYA patients who tend to be noncompliant.

Despite the recommendation of shorter treatment durations in AYA patients, some adjuvant chemotherapy, including chemotherapy agents other than those included in VAC or IVA, should be considered as a means of counteracting the possible cross-resistance of residual tumor cells posited by the CG hypothesis. Although a randomized clinical trial is warranted to determine the optimal combination of agents and duration of treatment for RMS, repeating small, randomized, phase 2 trials, possibly with a Bayesian design to screen for candidates for the phase 3 trial, might be helpful for developing a treatment, especially for AYA RMS.

### 4.4. VAC, Alternating VAC, and Other Regimens

Given the significantly high recurrence rate and poor prognosis of recurrent RMS, reserving alternative chemotherapy agents, such as (V)IE, VTC, VI, etc., in preparation for recurrences might be advisable. Among these treatments, VAC/VI demonstrated increased local failure and no survival benefit compared with VAC alone in the ARST0531 trial, possibly because the cyclophosphamide dosage was reduced, as previously discussed [16,38]. The COG is currently conducting ARST2031 to compare a VAC backbone with an intermediate cyclophosphamide dose (1.2 g/m^2^/cycle) and an experimental regimen containing vinorelbine, actinomycin-D, and cyclophosphamide [47]. Assessment of the feasibility, safety, and efficacy of this experimental regimen for newly diagnosed RMS, especially in AYA, await completion of the trial.

If a patient is treated with VAC chemotherapy only (still the standard RMS treatment), various other chemotherapy regimens may be held in reserve to treat recurrences. In a retrospective analysis by the International Society of Paediatric Oncology (SIOP), 474 patients with nonmetastatic RMS (including embryonal sarcoma) who experienced a recurrence had on average a 36% post-recurrence four-year OS rate [48]. Considering the potential curability of patients with recurrent RMS, having more treatment options in reserve for possible recurrences may improve the final treatment outcomes.

At present, however, there is no evidence to corroborate this view. Future development of novel agents may alter the current therapeutic scene and the decision-making process regarding treatment. The following two sections will discuss topics related to novel therapies.

### 4.5. Possible Re-Evaluation of Cytotoxic Chemotherapy Agents

The role of doxorubicin in RMS treatment has been controversial for more than three decades. Despite its significant activity against both newly diagnosed and recurrent RMS, the additive effect of doxorubicin in the standard VAC chemotherapy has not been demonstrated in either IRS-I or II [11,12]. After several decades, RMS2005 again addressed the same research question, but failed to prove the efficacy of additional doxorubicin [17]. An intensive, six-drug, combination chemotherapy with biweekly VDC-IE backbone in ARST0431 [34] demonstrated minimal efficacy and may not be suitable for AYA RMS as discussed above. Liposomal doxorubicin, a less toxic anthracycline derivative, is no longer used for sarcoma treatment, probably because of the low selectivity of its drug-delivery function for sarcomas other than the Kaposi sarcoma [49]. Our research group recently began investigating a novel agent known as polymer-conjugated pirarubicin (P-THP), which was developed for pediatric malignancies. P-THP, which targets tumor tissue highly selectively via an enhanced permeability and retention (EPR) effect, secondarily releases active pirarubicin molecules only in the acidic environment surrounding the tumor, but not in the vital organs [50].

Cisplatin, a conventional cytotoxic agent, is no longer used as a component of frontline therapy for RMS, but an objective response was observed in three of 16 children with recurrent RMS in a phase 2 trial of the drug [51]. As discussed above, cisplatin was clinically tested in IRS-III in regimens #35 and #36. Although the cisplatin-containing regimens demonstrated no survival benefit, they achieved a CR rate (57% in #35 and 62% in #36) superior to that of the VAC regimen (50% in #34) in clinical group IV [13]. Organ toxicities are inevitable for cisplatin and doxorubicin as the cumulative dose increases. The optimal dosage schedule and appropriate supportive care, such as adequate intravenous fluids and magnesium supplementation [52,53], is necessary when using cisplatin in a combination regimen.

### 4.6. Incorporating Molecularly Targeted Agents, Immunotherapy, and Cell Therapy

Since these treatments are relatively new, their application to RMS treatment is still in the research phase. COG compared the molecularly targeted agents, bevacizumab (BV) and temsirolimus (TEM), in a randomized phase 2 trial for patients with RMS with a first relapse having an unfavorable prognosis [54]. Eighty-seven patients (44 in the BV arm and 42 in the TEM arm; one ineligible) were enrolled. The six-month EFS rate for the BV arm was 54.6% (95% CI: 39.8% to 69.3%) and 69.1% (95% CI: 55.1% to 83%) for the TEM arm. The latter also demonstrated a superior response rate and fewer cases of progressive disease at six weeks. The effectiveness of TEM will be assessed in the final analysis of ARST1431, the current phase 3 trial, which is comparing a VAC/VI alternating regimen with an alternating VAC/VI plus TEM regimen in patients with newly diagnosed, intermediate risk RMS [55].

Although cell therapy using reduced-intensity allogeneic stem cell transplantation aiming for the graft-versus-tumor effect was not successful in treating RMS [56], there are several encouraging reports of the effectiveness of genetically engineered T-cell immunotherapy. In a phase 1/2 clinical trial of T cells expressing human epidermal growth factor receptor 2 (HER2)-specific chimeric antigen receptor (HER2 CAR T) in patients with a HER2-positive sarcoma, four of 17 patients achieved stable disease (SD) [57]. A 7-year-old patient with refractory bone marrow-metastatic RMS, who participated in a phase 1 clinical trial of HER2 CAR T cell immunotherapy (NCT00902044), achieved remission after three cycles of autologous HER2 CAR T cell immunotherapy following lymphodepleting chemotherapy and consolidation with additional four infusions of CAR T-cells without lymphodepletion [58]. Immune monitoring showed remodeling of the T-cell receptor repertoire with immunodominant clones and serum autoantibodies, which are reactive to oncogenic signaling pathway proteins. Although the disease relapsed in the bone marrow six months after stopping the therapy, another cycle of lymphodepletion and CAR T-cell infusion successfully induced a second remission. Although additional T-cell infusions and pembrolizumab were given following response consolidation with additional CAR, the patient has been off T-cell immunotherapy for 20 months without any signs of recurrence [58].

Two clinical trials tested immune checkpoint inhibitors in pediatric patients, including those with RMS. One phase 1 trial of nivolumab enrolled 85 patients, including 12 patients with RMS, none of whom demonstrated an objective response [59]. Another phase 1 trial using ipilimumab enrolled 33 patients, including two patients with RMS. Owing to the nature of the phase 1 trial and the small number of patients with RMS, efficacy assessment was not possible [60]. Despite these discouraging results, a case report of a 19-year-old patient with metastatic, chemotherapy-resistant, pleomorphic RMS [61] demonstrated encouraging treatment results. After whole exome sequencing of the patient’s tumor and germline DNA, a pathogenic, germline mutation (c.1863_1864insT) was found in the *MLH1* gene (p. Leu622Serfs*10), despite the absence of the classical criteria of Lynch syndrome in his familial history. The tumor was found to contain a loss of heterozygosity at the *MLH1* locus. Immunohistochemistry demonstrated *MLH1* and *PMS2* loss of nuclear expression. Given the mismatch repair defects and high expression of programmed cell death ligand 1 (PD-L1), the patient started nivolumab and achieved a rapid, complete response of the lung metastases which he has maintained past his one-year follow-up [61]. This case demonstrates that precision medicine may have benefits for AYA RMS with a genetic predisposition to cancer development.

### 4.7. Special Considerations Regarding Local Treatment

The timing and intensity of local treatments for patients with AYA RMS should be reappraised, especially because maintaining dose intensity in chemotherapy is difficult, as described in the previous chapter.

The prognostic value of the initial surgical resection demonstrated in a series of IRS studies corroborates the NS hypothesis explaining the relationship between local therapy and chemotherapy for RMS; the less residual disease there is, the greater the efficacy of later chemotherapeutic treatment is likely to be. In this regard, earlier local control, either via surgery or radiotherapy, in which the risk-benefit ratio must be carefully considered to choose the modality, should increase the possibility of a higher success rate for multimodal therapy. However, this proved not to be the case in ARST0531, which demonstrated an increased rate of local failure despite the induction of radiotherapy at week 4 [45]. Although the reason for the high failure rate is unclear, the slow responsiveness of RMS to chemotherapy and the intensity of the induction chemotherapy may be to blame. Optimizing induction chemotherapy in relation to the timing of radiotherapy warrants further investigation.

Unlike low-risk patients who have undergone complete resection of a small tumor in primary surgery, most patients with AYA RMS need radiotherapy. Since each AYA RMS case is treated with curative intent, the risk of radiotherapy, which increases the late effects of treatment, should not be underestimated. For example, the cumulative effect of radiation 2 Gy to the testes or 6–15 Gy to the ovaries can cause gonadal failure [62]. Moreover, the threshold levels are even lower in RMS because combination chemotherapy using high-dose alkylating agents is used.

The COG recommends modulating the local radiation dosage depending on the local disease status: 50.4 Gy for gross residual disease, 41.4 Gy for minimal residual disease, and 0–36 Gy for completely resected disease, although the dosages depend on the timing of radiation (before or after neoadjuvant chemotherapy) and the risk factors of the original disease [15,16]. Conversely, aggressive surgical resection of the tumor can allow a lower radiotherapy dosage for the same lesion. This view does not accord with the IRS surgical guidelines, which recommend minimal tumor resection to avoid damaging organ function and to minimize the effect on the patient’s physical appearance. However, striking an appropriate balance between aggressive surgical resection and radiotherapy is extremely important, not only for minimizing the radiation dosage but also for increasing the local control rate.

The IRS surgical guidelines recommend minimal tumor resection followed by radiotherapy for female patients with RMS of the reproductive organs. However, adding radiation targeting residual disease in the uterus may damage the hormonal function of the ovaries. Therefore, a simple hysterectomy without an oophorectomy, radiotherapy, or cyclophosphamide can be an option for uterine RMS if the lesion can be completely resected (Clinical Group I tumors in a favorable site can be treated with vincristine and actinomycin D only). A recently published expert consensus of the COG, the EpSSG, and the CWS supports this view, albeit under specific circumstances [63,64,65].

On the other hand, if we were to follow the suggestion that radical surgery be limited to cases in which organ function can be preserved, curative treatment of so-called “unresectable” disease might have to be given up and radiotherapy be administered only as the sole form of local treatment. However, this minimally surgical approach certainly carries a high risk of early local failure of the disease. Cases of RMS originating in an extremely rare site, such as the diaphragm, are often treated without radical surgery, but their time to progression is usually brief [66,67]. Our team recently experienced a case of primary diaphragmatic RMS in a teenaged patient who was successfully treated with aggressive, extensive surgery combined with radiation therapy and VAC-based chemotherapy [68]. Although preservation of organ function has the highest priority, the balance between surgical invasiveness and effectiveness should be carefully considered at least for a certain subpopulation of patients with RMS.

Of course, safe, minimally invasive surgery with the lowest possible risk of surgical complications should always be prioritized. To this end, sentinel lymph node mapping and biopsies, which are currently used to stage adult breast cancer and melanoma, may be applied to RMS in the extremities [69] or the head and neck region [70]. Radiotherapy for nodes with a suspected malignancy, but without histological confirmation via a biopsy, may lead to overtreatment. Once established, the technology of sentinel lymph node mapping will enable safer, more effective control of regional lymph nodes.

Last but not least, the necessity of minimizing the toxicity of radiotherapy should be considered. Intensity-modulated photon radiotherapy (IMRT) is widely used, especially for RMS in the head and neck region. Although the D9803 trial enrolling 375 patients demonstrated that IMRT had better target dose coverage than three-dimensional conformal radiotherapy, it failed to demonstrate any improvement in locoregional control or FFS [71]. Proton radiotherapy, which is currently being tested, is expected to have a more intensive treatment effect with fewer toxicities. Since the proton beam emits most of its energy near the end of the arrival depth, it can be highly concentrated at the target lesion to avoid damaging the surrounding, normal tissue, thus possibly reducing the incidence of radiation-related complications. In a phase 2 study of proton beam therapy in 57 patients with metastatic ERMS concomitantly treated with chemotherapy, the five-year EFS, OS, and local control rate was 69%, 78%, and 81%, respectively [72]. Despite some instances of acute, chronic, radiation-related toxicities (13% and 7% in this trial), proton beam therapy can be an option for pediatric and AYA patients with malignancies. Additional research is warranted to determine the safety and efficacy of proton beam therapy for RMS.

Each AYA patient with RMS should have the right to select a treatment strategy based on his/her own priorities. Therefore, a medical team consisting of specialists in each aspect of the strategy is needed to deliver a treatment optimized for individual AYA patients with RMS based on their expertise.

### 4.8. Psychosocial Issues in AYA RMS and the Perspective of Clinical Researchers

AYA patients experience various physiological and psychosocial changes which increase their fragility. Because they are at an age when the tendency for social independence becomes stronger, diagnostic delays may occur in the absence of adult supervision [73]. These patients also tend to be noncompliant to treatment and uncooperative. Young adults may not have medical insurance or have poor access to clinical trials [74] and may be especially susceptible to infertility or secondary malignancies resulting from treatment toxicities. Intensive countermeasures, including developing systems to provide specialized medical care, such as collaboration among pediatric oncologists, adult medical oncologists dealing with the same disease, and multidisciplinary teams comprised of surgeons and radiation oncologists [6]. Moreover, cancer survivors who received their diagnosis between ages 15 and 21 years are at increased risk of adverse mental health outcomes [75]. Therefore, mental health must be addressed along with treatment concerns, and specialized clinical care and survivorship programs addressing these various issues are necessary. In addition, a peer support system among AYA patients is often helpful in resolving a variety of psychosocial issues through the sharing of experiences. Our colleagues recently established a social network service in which AYA patients can communicate each other using the internet via a cellular phone or tablet [76].

Pediatric patients are often reluctant to accept a physician’s advice or instructions in a clinical trial, even if after they have received a thorough explanation about the treatment because their parents make most of their decisions for them. On the other hand, AYA patients possess enough knowledge and common sense to understand the challenges involved in clinical trials, such as possible treatment toxicities and issues related to their prognosis. Compared to older adults, they more readily agree with the physician’s choice of treatment than with the trial treatment, which is often accompanied by numerous restrictions, because they are relatively free of the sense of social obligation, etc. which influence the behavior of adults. By the same token, they may easily drop out of a trial if they feel the procedure has no benefit to them or find the treatment to be too toxic in the short term. Considering the low treatment compliance of AYA patients, researchers should try to explain the objectives of a clinical trial clearly, emphasizing its possible benefits to encourage greater compliance and cooperation in AYA patients.

The Declaration of Helsinki [77] states that “the responsibility for the protection of research subjects always lies with the physician or other health care professionals and never with the research subjects themselves even if they have given their consent” for participation. Although sequential clinical trials for AYA RMS are the only way to improve treatment outcomes in these patients, physicians should always try to find a therapeutic approach optimized for the individual and carefully compare the risks and benefits of this approach with those of the clinical trial before asking the patient to choose. Although the best therapeutic approach for AYA RMS remains elusive, the previous studies mentioned in this article offer some hints. In the final analysis, the care taken by the physician and researcher to help patients explore the various, available options is the best ally of AYA RMS patients.

## 5. Conclusions

A proverb in the Analects of Confucius exhorts us to “visit the old, learn the new.” In the present review, the author has attempted to combine his own insights, gleaned from data from past international clinical trials, with a brief consideration of new findings regarding novel therapies with possible applications to AYA RMS. Organizing a holistic treatment approach optimized for individual patients is possible by creatively integrating new findings, such as gene-based diagnostics and treatments, with what we have learned so far from basic and clinical research.

## Figures and Tables

**Figure 1 cancers-14-02270-f001:**
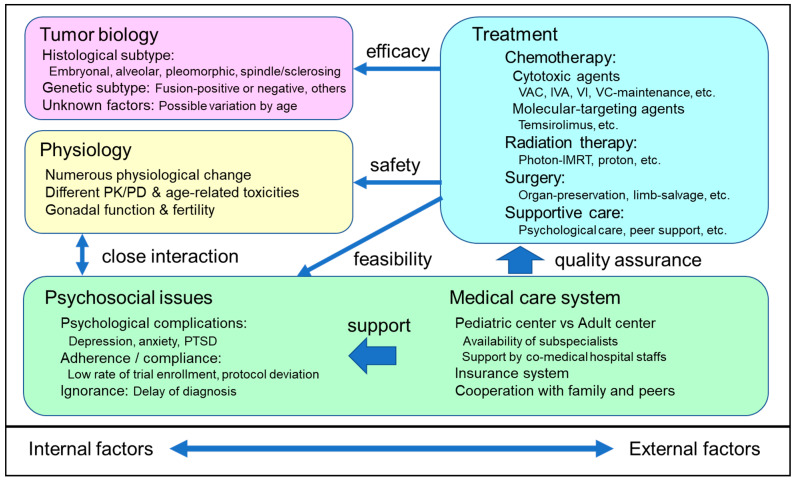
Major factors influencing the poor outcomes in AYA RMS. The reason for the poor RMS outcomes in AYA patients is multifactorial. The factors significantly influencing poor outcomes include internal factors, external factors, and a combination of various issues. The impact of each factor will be discussed in the following chapters. IMRT, intensity-modulated photon radiotherapy; IVA, ifosfamide, vincristine, actinomycin-D; PD, pharmacodynamics; PK, pharmacokinetics; PTSD, post-traumatic stress disorders; VAC, vincristine, actinomycin-D, cyclophosphamide; VC, vinorelbine, cyclophosphamide; VI, vincristine, irinotecan.

**Figure 2 cancers-14-02270-f002:**
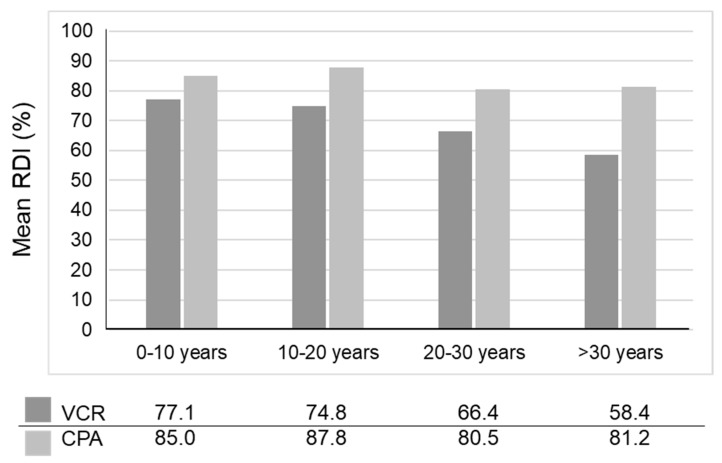
Relative dose intensity of vincristine and cyclophosphamide in relation to age. The relative dose-intensity of vincristine and cyclophosphamide tended to decrease as the patients aged. CPA, cyclophosphamide; RDI, relative dose intensity; VCR, vincristine. This figure was used with the permission of Kojima et al. [35].

**Figure 3 cancers-14-02270-f003:**
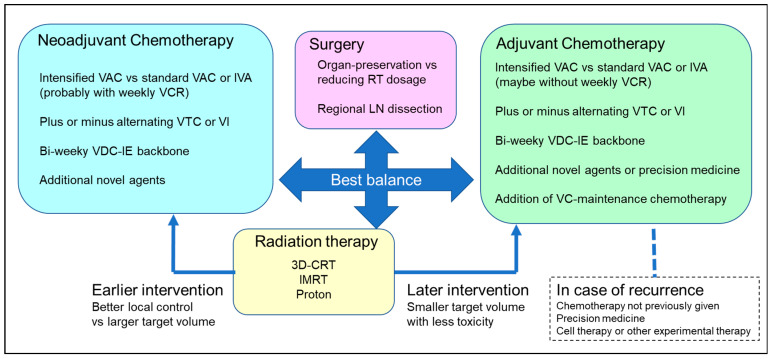
Basic structure of multimodal therapy for RMS and key clinical issues. Although each clinical question should be answered by conducting a well-designed clinical trial, the physician is responsible for selecting the optimal strategy in the real-world setting. 3D-CRT, three-dimensional conformal radiotherapy; IMRT, intensity-modulated photon radiotherapy; IE, ifosfamide, etoposide; IVA, ifosfamide, vincristine, actinomycin-D; LN, lymph node(s); RT, radiotherapy; VAC, vincristine, actinomycin-D, cyclophosphamide; VC, vinorelbine, cyclophosphamide; VCR, vincristine; VDC, vincristine, doxorubicin, cyclophosphamide; VI, vincristine, irinotecan; VTC, vincristine, topotecan, cyclophosphamide.

**Table 1 cancers-14-02270-t001:** Comparison of the mean relative dose-intensity between adults and children.

Drug	Treatment Phase	Adult (%)	Child (%)	*p*-Value
Vincristine	Induction phase	77.7	86.1	0.109
	Maintenance phase	71.9	100.1	0.042
	Total phase	76.8	90.2	0.040
Cyclophosphamide	Induction phase	87.1	88.2	0.820
	Maintenance phase	69.7	86.4	0.011
	Total phase	80.0	86.9	0.156

**Table 2 cancers-14-02270-t002:** Chemotherapy regimens available for newly diagnosed RMS.

Regimen	Trial	Dosage (mg/m^2^) and Schedule	Ref.
VAC	IRS-IV	VCR 1.5 on days 1, 8, 15; ACD 0.015/kg on days 1–5; CPA 2200 on day1; every 3 weeks	[14]
VAC	D9802/ D9803	VCR 1.5 on days 1, 8, 15; ACD 1.5 on day 1; CPA 2200 on day1; every 3 weeks	[15,37]
VAC	ARST0531	VCR 1.5 on days 1, 8, 15; ACD 1.5 on day 1; CPA 1200 on day 1; every 3 weeks	[16]
VIE	IRS-IV	VCR 1.5 on days 1, 8, 15; IFM 1800 on days 1–5; ETP 100 on days 1–5; every 3 weeks	[14]
VAI	IRS-IV	VCR 1.5 on days 1, 8, 15; ACD 1.5 on day 1; IFM 1800 on days 1–5; every 3 weeks	[14]
VTC	D9803	VCR 1.5 on days 1, 8, 15; Topo 250 on days 1–5; CPA 250 on days 1–5; every 3 weeks	[15]
VI	ARST0431/ ARST0531	VCR 1.5 on days 1, 8, 15; IRI 50 on days 1–5; every 3 weeks	[16,38]
VDC	ARST0431	VCR 1.5 on days 1, 8, 15; DXR 37.5 on days 1, 2; CPA 1200 on day 1; every 2 weeks alternating with IE	[39]
IE	ARST0431	IFM 1800 on days 1–5; ETP 100 on days 1–5; every 2 weeks alternating with VDC	[39]
IVA	RMS2005	IFM 3000 on days 1–2; VCR 1.5 on days 1, 8, 15; ACD 1.5 on day 1; every 3 weeks	[17]
VC maintenance	RMS2005	VNR 25 on days 1, 8, 15; CPA (po) 25 daily; for 4 weeks cycles × 6 cycles	[18]

ACD, actinomycin-D; CPA, cyclophosphamide; DXR, doxorubicin; ETP, etoposide; IFM, ifosfamide; IRI, irinotecan; IE, ifosfamide, etoposide; IVA, ifosfamide, vincristine, actinomycin-D; Topo, topotecan; VAC, vincristine, actinomycin-D, cyclophosphamide; VC, vinorelbine, cyclophosphamide; VCR, vincristine; VDC, vincristine, doxorubicin, cyclophosphamide; VI, vincristine, irinotecan; VIE, vincristine, ifosfamide, etoposide; VNR, vinorelbine; VTC, vincristine, topotecan, cyclophosphamide.

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
