# Peer review of "Optimizing Rhabdomyosarcoma Treatment in Adolescents and Young Adults"

_cancers, 2022, doi:10.3390/cancers14092270_

Round 1

Reviewer 1 Report

Overall, the review touches different topics and aspects in the management of adolescents and young  adults with rhabdomyosarcoma. The language needs some review given the fact some sentences appear a bit convoluted and/or unclear. Given the amount of data from clinical trials, it is sometimes difficult to follow the discussion, maybe adding a summary table with the main prospective studies can help. Additional comments below.

-line 52: adult patients cut off? age 18?

-IVA regimen also mentioned as VAI: use the same abbreviation consistently

-line 200: neoadjuvant chemotherapy is not only aimed to improve local tumor control but also work on micrometastatic disease

-line 215: well-designed clinical trials do not only require considerable time but also financial effort and coordination among different centers in potentially different countries with different policies

-lines 337-339: phase 2 randomized trials with a bayesian design should also be considered

-line 384: these drugs accumulate? You mean, as the cumulative dose increases?

Author Response

Thank you for your evaluation and valuable comments. Following your recommendation, I added Table 2, which summarizes the chemotherapy regimens used in the clinical trials cited in this article. In addition, I have had a professional English editor thoroughly review the manuscript prior to re-submission. Answers for each comment are given below.

-line 52: adult patients cut off? age 18?

Answer: Cut off value is age 19. I added the number in line 51.

-IVA regimen also mentioned as VAI: use the same abbreviation consistently

Answer: Although IVA and VAI share the same drugs, the dosages and schedules are different. I added the information in Table 2, which is newly added.

-line 200: neoadjuvant chemotherapy is not only aimed to improve local tumor control but also work on micrometastatic disease

Answer: Yes. I added “mainly” in line 211 (in the revised manuscript).

-line 215: well-designed clinical trials do not only require considerable time but also financial effort and coordination among different centers in potentially different countries with different policies

Answer: Your suggestion is important. I added the information about the financial issues and difficulties with collaboration.

-lines 337-339: phase 2 randomized trials with a bayesian design should also be considered

Answer: Yes. I added the phrase “possibly with a Bayesian design” in line 355 (in the revised manuscript).

-line 384: these drugs accumulate? You mean, as the cumulative dose increases?

Answer: Thank you for your advice. I changed the phrase from “as these drugs accumulate” to “as the cumulative dose increases”.

Reviewer 2 Report

This is a review article on the topic of treatment of rhabdomyosarcoma in adolescence and young adults. In general, the manuscript is well written and seems to attract the readers of the journal. I recommend it for publication after some minor revision. 

Followings are specific comments.

1. The authors mentioned gene-based diagnostics and treatments but doesn’t describe well in the manuscript. It’s worth introducing these new strategies as well as ongoing basic research.

2. The authors specifically described the surgical treatment for tumors in the reproductive organs. It would be better to discuss more the differences of the local treatment according to the location of the tumor.

3. Adding tables or schemas to summarize each regimen would help readers understand easily.

4. The last paragraph 4.8 is too general and seems irrelevant to the topic.

Author Response

Thank you for your evaluation and valuable comments. Answers for each comment are given below.

  1. The authors mentioned gene-based diagnostics and treatments but doesn’t describe well in the manuscript. It’s worth introducing these new strategies as well as ongoing basic research.

Answer: I believe that gene-based diagnostics are described in “2. Tumor biology” and examples of treatment strategy based on genes and molecular information are described in “4.6. Incorporating molecularly targeted agents, immunotherapy, and cell therapy “.

  1. The authors specifically described the surgical treatment for tumors in the reproductive organs. It would be better to discuss more the differences of the local treatment according to the location of the tumor.

Answer: The description of surgical treatment covers the major sites of rhabdomyosarcomas, including head & neck, urogenital, and extremities. Because the principle of surgery is basically same as that for pediatric rhabdomyosarcoma, there are not many differences between pediatric patients and AYA patients to discuss here. Therefore, I just described “special consideration” for surgery in AYA patients.

  1. Adding tables or schemas to summarize each regimen would help readers understand easily.

Answer: I agree and added Table 2 summarizing the regimens.

  1. The last paragraph 4.8 is too general and seems irrelevant to the topic.

Although the paragraph contains general information, the information here is quite important for discussing AYA RMS. It is relevant to the topic.

Reviewer 3 Report

The theme of the review is very important. AYA is a group of patients who are managed by both paediatric oncologists and internal oncologists. Unfortunately, paediatric and internal oncology have different treatment methods and ways of dealing with patients. Often studies for RMS include only patients up to 18. So it is also a problem of the organisation of the health care system and the responsibility of the disciplines (different from country to country) that AYA patients are treated particularly badly. 

Special comments:

Figures 1 –

This figure does not show the causes for the poorer prognosis in adults, but rather general factors that can influence the outcome.  It randomly lists things like organ preservation or an IVA , IV without any reference what for. This figure needs to be either fundamentally rewritten or deleted.

69

“RMS refers to malignant tumors with rhabdomyoblastic differentiation”

Not all RMS show rhabdomyoblastic differentiation.

84

“differ significantly among these subtypes, the treatment strategy is the same”.

 Not correct RMS subtype is a major risk factor and is used in risk stratification and influences therapy

126

Moreover, a genomic characterization study of MSKCC (MSK-IMPACT) revealed that TP53 mutations, which may predict  RMS! Probably   something is missing in this sentence (poor prognosis ?)

180

“intolerance to the treatment but also the risk of infertility in this age group”

 Not exactly true. Gonadal toxicity is age-independent and even more severe in younger patients (pre - puberty), as cryopreservation of sperm or testicular tissue is not possible or very complicated in very young children

 203

“In contrast, adjuvant chemotherapy, which is administered in the presence of minimal residual disease”

Neoadjuvant CHT is also effective against MRD, which also exists in the preoperative therapy phase.

207-208

A certain amount of adjuvant chemotherapy is necessary to eradicate the residual disease in RMS patients even  when the best possible combination of agents is used.”

It is not clear what the author wants to tell us

Figure 3

Figure 3 is also not good - it is partly a repetition of figure 1, it is not very informative and does little to improve the content of the article. Basically remodel or paint.

235-237

“The following discussion will hopefully aid readers in optimizing treatments for 235 patients not only with AYA RMS but also childhood RMS. The author should discuss the  results and how they can be interpreted from the perspective of previous studies and of 237 the working hypotheses.”

To whom is this recommendation addressed?

257-258

“These data may have discouraged researchers from trying to optimize neoadjuvant chemotherapy to improve tumor resectability and the local control rate”

Publications of other European group CWS on the issue of response should be discussed and cited. (Dantonello et al. PBC 2015, Dantonello et al. JCO 2009)

 263-267

“However, devising superior drug combinations and treatment schedules for neoadjuvant chemotherapy may contribute to improving tumor resectability and the local control rate. As well, superior drug combinations and dosage schedules for reducing systemic 265 minimal residual disease should be feasible with adjuvant chemotherapy”

What does the author mean by "superior drug combination" The problem with RMS therapy is that despite many randomised trials, no benefit has been seen from other drugs in combination with VAC or IVA.

Additionally, publications of other European group (CWS) on the issue of VCR dose intensification (weekly  preoperative)  should be discussed  and cited ( Koscielniak et al. Cancers 2022).

271-273

EpSSG and CWS never used the VCR weekly during the whole therapy.

Therefore, it does not have to be omitted in AYA.

319-321

“The RMS2005 trial conducted by the EpSSG demonstrated a clear benefit to adding a 24-week maintenance chemotherapy regimen consisting of vinorelbine and oral cyclo-320 phosphamide in term of both EFS and OS for non-metastatic high-risk RMS”

Wrong statement!! Only OS was significantly better in the MT arm! But NOT EFS!

442-443

“In this regard, earlier local control either via surgery or radiotherapy should achieve a higher success rate in multimodal therapy”

It is not always true. Resection attempts resulting in tumor debulking and tumor spread may deteriorate prognosis. There is a need to warn against ill-considered surgical interventions.

The risk-adapted determination of the sequence of surgery and radiotherapy, or the choice of only one local measure, is existentially important, especially in view of the diversity of localization in RMS.

475

Other Instruct consents on 1.Paratesticular RMS Rogers et al. PBC 2021, and 2. Extremity RMS Morris et al. BBC 2021 should be cited.

477

“curative treatment of so-called “unresectable” disease might have to be given up and radiotherapy might be administered only as a form of local treatment.”

Two comments:

 Language: the author means probably “as the sole form of local treatment”

Content: RT as a sole form of local treatment is indicated  in primary sites i.e. head neck parameningeal, where even mutilating surgery does not results in R0 resection.

485

“Rather than placing too much emphasis on preserving organ function and thereby compromising curability, a well-organized, multidisciplinary strategy employing curative surgery is recommended at least for a certain subpopulation of patients with RMS”.

This sentence should be deleted or re-write. Preservation of organ function has the highest priority in young patients who should not only survive but also have a good quality of life. Only if there is no other option must mutilating surgery be performed.

 506

“Despite some instances of acute, chronic, radiation-related toxicities (13% and 7% in this trial), proton beam therapy is considered beneficial for pediatric and AYA patients with malignancies”

That is not true. To date, there is no evidence that proton therapy is beneficial in comparison to photons, either for local control or for minimizing late effects.

511

“Each AYA patient with RMS should have the right to select a treatment strategy  based on his/her own priorities”

Every patient has the right to agree or refuse therapy, not just AYA. However, the hospital is not a self-service shop where you can choose a few things. The patient must be able to rely on the competence of the doctors.

Author Response

Reply to Reviewer 3

Thank you very much for the many valuable comments and recommendations, which have should help me to improve the quality of my article and eliminate errors in the descriptions. I agree with most of your comments and made changes accordingly. I did not agree with several of your points but offered an explanation of what I intend to say.

Figures 1 –

This figure does not show the causes for the poorer prognosis in adults, but rather general factors that can influence the outcome.  It randomly lists things like organ preservation or an IVA, IV without any reference what for. This figure needs to be either fundamentally rewritten or deleted.

Answer: I changed the figure title to “Major factors possibly influence the poor outcomes in AYA RMS” to avoid misunderstanding. I believe that this figure is important to guide readers in the following discussion. Therefore, I added “Impact of each factor will be discussed in the following chapters” in the legend.

69

“RMS refers to malignant tumors with rhabdomyoblastic differentiation”

Not all RMS show rhabdomyoblastic differentiation.

Answer: I changed “RMS refers to malignant tumors with skeletal muscle differentiation“, which I quoted from the 2020 WHO classification of soft tissue tumors. (line 71 in the revised manuscript)

84

“differ significantly among these subtypes, the treatment strategy is the same”.

 Not correct RMS subtype is a major risk factor and is used in risk stratification and influences therapy

Answer: I changed the part to “differ significantly among these subtypes, both share the same chemotherapeutic regimens”. (line 86 in the revised manuscript)

126

Moreover, a genomic characterization study of MSKCC (MSK-IMPACT) revealed that TP53 mutations, which may predict RMS! Probably   something is missing in this sentence (poor prognosis?)

Answer: This is my mistake. As you point out, it means “Moreover, a genomic characterization study of MSKCC (MSK-IMPACT) revealed that TP53 mutations, which may predict poor prognosis of RMS”. (lines 128-129 in the revised manuscript)

180

“intolerance to the treatment but also the risk of infertility in this age group”

  Not exactly true. Gonadal toxicity is age-independent and even more severe in younger patients (pre - puberty), as cryopreservation of sperm or testicular tissue is not possible or very complicated in very young children

Answer: I understand your point but do not think that modification is necessary in this specific context.

 203

“In contrast, adjuvant chemotherapy, which is administered in the presence of minimal residual disease”

Neoadjuvant CHT is also effective against MRD, which also exists in the preoperative therapy phase.

Answer: I changed the phrase from “in the presence of” to “, which specifically targets” in the sentence. (line 214 in the revised manuscript)

207-208

A certain amount of adjuvant chemotherapy is necessary to eradicate the residual disease in RMS patients even when the best possible combination of agents is used.”

It is not clear what the author wants to tell us

Answer: I changed the sentence to “A certain dosage and duration of adjuvant chemotherapy is necessary to eradicate the residual disease in RMS patients even when the best possible combination of agents is used.” (lines 217 – 219 in the revised manuscript)

Figure 3

Figure 3 is also not good - it is partly a repetition of figure 1, it is not very informative and does little to improve the content of the article. Basically remodel or paint.

Answer: I understand that the content of Figure 3 is too basic and unnecessary for specialists like you. But I believe this would help readers with other oncology backgrounds to understand the following discussion.

235-237

“The following discussion will hopefully aid readers in optimizing treatments for patients not only with AYA RMS but also childhood RMS. The author should discuss the  results and how they can be interpreted from the perspective of previous studies and of 237 the working hypotheses.”

To whom is this recommendation addressed?

Answer: I do not think this part is necessary and therefore removed these sentences.

257-258

“These data may have discouraged researchers from trying to optimize neoadjuvant chemotherapy to improve tumor resectability and the local control rate”

Publications of other European group CWS on the issue of response should be discussed and cited. (Dantonello et al. PBC 2015, Dantonello et al. JCO 2009)

Answer: I added the discussion (in lines 265 – 273) and these references (as #39 and #40).

 263-267

“However, devising superior drug combinations and treatment schedules for neoadjuvant chemotherapy may contribute to improving tumor resectability and the local control rate. As well, superior drug combinations and dosage schedules for reducing systemic minimal residual disease should be feasible with adjuvant chemotherapy”

What does the author mean by "superior drug combination" The problem with RMS therapy is that despite many randomised trials, no benefit has been seen from other drugs in combination with VAC or IVA.

Answer: What I meant here was not the clinical trials (for the mass) but the individual patient. We can judge the effectiveness of each treatment regimen. If a certain regimen is ineffective, we might select another one which possibly works better.

Additionally, publications of other European group (CWS) on the issue of VCR dose intensification (weekly preoperative) should be discussed and cited ( Koscielniak et al. Cancers 2022).

Answer: I added the discussion (in line 284) and these references (as #43).

271-273

EpSSG and CWS never used the VCR weekly during the whole therapy.

Therefore, it does not have to be omitted in AYA.

Answer: I changed the sentence to “The continuation phase of treatment without weekly vincristine to avoid excessive neurotoxicity is one way of optimizing AYA RMS treatment”. (in lines 288 – 290)

319-321

“The RMS2005 trial conducted by the EpSSG demonstrated a clear benefit to adding a 24-week maintenance chemotherapy regimen consisting of vinorelbine and oral cyclophosphamide in term of both EFS and OS for non-metastatic high-risk RMS”

Wrong statement!! Only OS was significantly better in the MT arm! But NOT EFS!

Answer: I changed the sentence to “The RMS2005 trial conducted by the EpSSG demonstrated a benefit of adding a 24-week maintenance chemotherapy regimen consisting of vinorelbine and oral cyclophosphamide in term of OS for non-metastatic high-risk RMS.” (in lines 336 – 338)

442-443

“In this regard, earlier local control either via surgery or radiotherapy should achieve a higher success rate in multimodal therapy”

It is not always true. Resection attempts resulting in tumor debulking and tumor spread may deteriorate prognosis. There is a need to warn against ill-considered surgical interventions.

The risk-adapted determination of the sequence of surgery and radiotherapy, or the choice of only one local measure, is existentially important, especially in view of the diversity of localization in RMS.

Answer: Following your advice, I changed the sentence to “In this regard, earlier local control either via surgery or radiotherapy, in which the risk-benefit ratio must be carefully considered to choose the modality, should increase a possibility of higher success rate in multimodal therapy.” (in lines 460 – 463)

475

Other Instruct consents on 1.Paratesticular RMS Rogers et al. PBC 2021, and 2. Extremity RMS Morris et al. BBC 2021 should be cited.

Answer: I added these references (as #64 and #65).

477

“curative treatment of so-called “unresectable” disease might have to be given up and radiotherapy might be administered only as a form of local treatment.”

Two comments:

 Language: the author means probably “as the sole form of local treatment”

Content: RT as a sole form of local treatment is indicated in primary sites i.e. head neck parameningeal, where even mutilating surgery does not results in R0 resection.

Answer: I changed the sentence to “curative treatment of so-called “unresectable” disease might have to be given up and radiotherapy might be administered only as the sole form of local treatment.” (in lines 496 – 498)

485

“Rather than placing too much emphasis on preserving organ function and thereby compromising curability, a well-organized, multidisciplinary strategy employing curative surgery is recommended at least for a certain subpopulation of patients with RMS”.

This sentence should be deleted or re-write. Preservation of organ function has the highest priority in young patients who should not only survive but also have a good quality of life. Only if there is no other option must mutilating surgery be performed.

Answer: I changed the sentence to “Although preservation of organ function has the highest priority, the balance between surgical invasiveness and effectiveness should be carefully considered at least for a certain subpopulation of patients with RMS.” (in lines 504 – 506)

 506

“Despite some instances of acute, chronic, radiation-related toxicities (13% and 7% in this trial), proton beam therapy is considered beneficial for pediatric and AYA patients with malignancies”

That is not true. To date, there is no evidence that proton therapy is beneficial in comparison to photons, either for local control or for minimizing late effects.

Answer: I changed the sentence to “Despite some instances of acute, chronic, radiation-related toxicities (13% and 7% in this trial), proton beam therapy can be an option for pediatric and AYA patients with malignancies.” (in lines 525 – 527)

511

“Each AYA patient with RMS should have the right to select a treatment strategy based on his/her own priorities”

Every patient has the right to agree or refuse therapy, not just AYA. However, the hospital is not a self-service shop where you can choose a few things. The patient must be able to rely on the competence of the doctors.

I added one phrase “based on their expertise” at the end of the paragraph. (in line 533)